# Endoplasmic Reticulum Stress in Acute Myeloid Leukemia: Pathogenesis, Prognostic Implications, and Therapeutic Strategies

**DOI:** 10.3390/ijms26073092

**Published:** 2025-03-27

**Authors:** Wojciech Wiese, Grzegorz Galita, Natalia Siwecka, Wioletta Rozpędek-Kamińska, Artur Slupianek, Ireneusz Majsterek

**Affiliations:** 1Department of Clinical Chemistry and Biochemistry, Medical University of Lodz, Mazowiecka 5, 92-215 Lodz, Poland; wojciech.wiese@stud.umed.lodz.pl (W.W.); grzegorz.galita@umed.lodz.pl (G.G.); natalia.siwecka@stud.umed.lodz.pl (N.S.); wioletta.rozpedek@umed.lodz.pl (W.R.-K.); 2Office of the Vice President for Research, Temple University, Philadelphia, PA 19140, USA

**Keywords:** acute myeloid leukemia, endoplasmic reticulum stress, unfolded protein response, hematopoietic stem cells, IRE1α, PERK, ATF6

## Abstract

Acute myeloid leukemia (AML) is a heterogeneous hematological malignancy that poses a significant therapeutic challenge due to its high recurrence rate and demanding treatment regimens. Increasing evidence suggests that endoplasmic reticulum (ER) stress and downstream activation of the unfolded protein response (UPR) pathway play a key role in the pathogenesis of AML. ER stress is triggered by the accumulation of misfolded or unfolded proteins within the ER. This causes activation of the UPR to restore cellular homeostasis. However, the UPR can shift from promoting survival to inducing apoptosis under prolonged or excessive stress conditions. AML cells can manipulate the UPR pathway to evade apoptosis, promoting tumor progression and resistance against various therapeutic strategies. This review provides the current knowledge on ER stress in AML and its prognostic and therapeutic implications.

## 1. Introduction

Acute myeloid leukemia (AML) is a heterogeneous hematologic malignancy characterized by abnormal clonal proliferation and differentiation of hematopoietic stem cells (HSCs). The 2022 WHO classification based on key mutations and genetic alterations distinguishes 32 subtypes of AML [1]. AML represents nearly 80% of cases of leukemia in individuals over 60 years old [2]. Despite advancements in AML treatments, managing this aggressive malignancy remains challenging. Overall survival rates are significantly affected by the persistence of measurable residual disease (MRD), which often leads to relapse [3]. These challenges underscore the need for novel therapeutic targets and reliable biomarkers to enhance treatment efficacy and improve patient prognosis.

The pathogenesis of AML involves chromosomal rearrangements, gene mutations, inflammation, epigenetic reprogramming, and deregulated cellular metabolism. The most common risk factors for developing AML include previous exposure to chemotherapy or radiotherapy, toxic substances, and a history of hematologic malignancies. Secondary AML represents 10% to 30% of all AML cases. The likelihood of developing AML rises with age, with a median onset of 68 years. It is commonly observed among men, Caucasians, and Pacific Islanders/Alaskan natives [4,5]. One of the most frequent mutations in newly diagnosed AML patients is the internal tandem duplication (ITD) of the FMS-like receptor tyrosine kinase 3 (FLT3) gene. Almost 30% of AML cases present this alteration and are linked to a rapidly progressing disease with a poor prognosis [6,7]. Mutations involving NPM, FLT3, and RAS typically are secondary events. Common genetic mutations include alterations in genes encoding epigenetic modifiers, such as DNMT3A, ASXL1, TET2, and IDH1. Alterations also occur in genes such as TP53, RUNX1, ASXL1, and CCAAT/enhancer-binding protein (CEBPA). Additionally, notable chromosomal translocations such as t(8;21), inv(16), and t(15;17) can contribute to the upregulation of JUN, which in turn enhances the expression of effectors involved in the unfolded protein response (UPR) [8,9,10,11].

Recent studies have drawn attention to the critical role of endoplasmic reticulum (ER) stress in AML’s development and progression [12,13]. The ER is an essential organelle in eukaryotic cells. When proteins misfold or accumulate within the ER, a condition known as ER stress occurs. To restore balance, cells activate an adaptive mechanism known as the UPR. However, the UPR may lead to apoptotic cell death if the stress is too severe. The activation of the UPR and ER stress has been associated with the development of several cancers, including colorectal cancer, lung cancer, multiple myeloma (MM), chronic lymphocytic leukemia (CLL), and AML [14,15,16,17,18]. Chronic ER stress and prolonged UPR activation are also associated with diseases such as neurodegenerative disorders, diabetes, autoimmune diseases, and cardiovascular diseases [19,20]. In the context of AML, UPR, and ER stress act as both a double-edged sword and a critical modulator of the disease, playing a significant role in leukemogenesis [21]. Moreover, targeting ER stress in AML has emerged as a promising strategy for overcoming resistance and improving treatment outcomes. AML cells often exhibit ER stress reactions that may hinder treatments’ success [22,23]. Moreover, ER stress can be a prognostic biomarker in AML [24]. Thus, this review presents the current knowledge we have gathered on the multiple roles of ER stress in AML.

## 2. Pathogenesis of ER Stress and the UPR

The ER plays many essential roles in the cell, including synthesizing, folding, and structurally maturing proteins, storing intracellular calcium ions, synthesizing and storing lipids, and metabolizing glucose. Disruptions in protein folding and modification can accumulate misfolded or unfolded proteins in the ER lumen, resulting in ER stress. This stress triggers the activation of the unfolded protein response (UPR) signaling pathway. The UPR is an adaptive response that promotes cell survival or activates apoptosis when homeostasis is not maintained [25]. The UPR consists of three primary signaling branches: inositol-requiring enzyme-1α (IRE1α), PKR-like ER kinase (PERK), and activating transcription factor-6 (ATF6) (Figure 1). There are many studies indicating that ATF6 is an important factor in the pathogenesis of diseases such as achromatopsia [26], Crohn’s disease [27], Parkinson’s disease [28], fatty liver disease [29], colon cancer [30], and type 2 diabetes mellitus [31]. The IRE1α- X-box-binding protein 1 (XBP1) pathway is the most conserved branch of the UPR. IRE1 is the oldest UPR sensor, first identified in yeast Saccharomyces cerevisiae [32]. The spliced version of XBP1, known as XBP1s, regulates the upregulation of a broad range of UPR-related genes. These genes are involved in various processes of ER-associated degradation (ERAD) [33]. ER stress causes caspase-mediated cleavage of IRE1. This process generates a stable fragment of IRE1 that includes an ER-lumenal domain and a transmembrane segment. The cleavage separates the stress-sensing and signaling portions of IRE1, reducing its activity [34]. PERK is an ER transmembrane receptor that remains inactive while associated with binding immunoglobulin proteins (BiP) chaperones. When unfolded or misfolded proteins accumulate in the ER lumen due to stress, BiP dissociates from PERK, leading to its activation through oligomerization and trans-autophosphorylation. This active kinase phosphorylates eIF2α, inhibiting global protein translation and causing a G1 phase cell cycle arrest. It also preferentially translates ATF4 to help restore cell homeostasis (Figure 1). If ER stress persists and the pro-adaptive UPR fails, PERK may activate pro-apoptotic signals via CHOP, promoting apoptosis [35]. ATF4 regulates genes essential for stress adaptation, but during prolonged ER stress, it can also activate CCAAT-enhancer-binding protein homologous protein (CHOP) genes, triggering apoptosis [36,37]. The UPR also becomes activated during hypoxia, infection, altered calcium homeostasis, abnormalities in lipid metabolism, and oxidative stress [38,39,40]. ATF6 moves from the ER to the Golgi apparatus, where it is cleaved by site-1 protease (S1P) and site-2 protease (S2P). After processing in the Golgi apparatus, ATF6 has the nuclear form of ATF6 (ATF6n). ATF6n migrates to the nucleus, activating transcription of the UPR target genes involved in ER protein folding homeostasis and cell physiology [41,42].

### ER Stress and the UPR in Bone Marrow

HSCs are the foundation of the hematopoietic system, playing a crucial role in the long-term maintenance and production of all mature blood cell lineages throughout an organism’s lifespan [43,44]. ER stress and UPR help HSCs maintain their self-renewal, multipotency, and survival. The knockdown of IRE1 in the hematopoietic compartment (Vav1-Cre) led to a partial impairment in the repopulation capacity of HSCs [45,46,47]. Estrogen therapy augments the regenerative capacity of HSCs following transplantation and accelerates hematopoietic recovery post-irradiation. Estrogen signaling through estrogen receptor α (ERα) in hematopoietic cells activates the protective Ire1α-Xbp1 branch of the UPR. Additionally, ERα-driven activation of the Ire1α-Xbp1 pathway enhances HSC resistance to proteotoxic stress and facilitates regenerative capacity [48]. Activating the UPR pathway blocks myeloid differentiation and deregulates the cell cycle, which are key features of the leukemic phenotype. This occurs by activation of calreticulin in the ATF6 pathway, therefore inhibiting CCAAT/enhancer-binding factor alpha (CEBPA), a protein essential for myeloid differentiation [49].

HSCs are exposed to harsh conditions in the bone marrow. The bone marrow niche has low oxygen concentrations, making hematopoietic stem and progenitor cells (HSPCs) susceptible to hypoxia, which can further increase ER stress. In low-oxygen environments, hypoxia-inducible factors (HIFs) activate, mediating a significant adaptive response to hypoxia that involves multiple pathways necessary for maintaining cellular homeostasis. It has been reported that HIF-2α-deficient HSPCs show higher levels of reactive oxygen species (ROS) generation (Figure 2). Elevated ROS can lead to ER stress and trigger apoptosis by activating the UPR. Additionally, ROS can cause genomic instability by damaging DNA, which may contribute to chemotherapy resistance and AML progression [50,51,52]. Specific adaptive pathways of UPR, such as the IRE1α-XBP1 pathway, help protect HSCs from apoptosis caused by ER stress, whereas an IRE1α knockout results in decreased reconstitution of HSCs [46]. Conversely, increased activation of PERK in HSCs makes them more sensitive to ER stress [45]. In vitro studies show that IL-4 induces apoptosis in HSCs and downregulates transcription factors involved in megakaryocyte development. IL-4Rαhigh HSCs are particularly sensitive to ER stress-induced apoptosis [53].

Moreover, research indicates that a deficiency of UFBP1, a vital component of the Ufm1 conjugation system essential for hematopoiesis and erythroid differentiation, leads to increased ER stress and activation of the UPR [54]. Depletion of Uba5 also results in heightened ER stress, reduced levels of the erythroid transcription factors GATA-1 and KLF1, and the blockage of erythroid differentiation from colony-forming unit-erythroid to proerythroblasts [54]. In contrast, the knockdown of activating signal cointegrator 1 (ASC1) affects the expression of transcription factors without directly impacting ER stress. ASC1 is a subunit of the ASC-1 complex, a transcriptional coactivator that regulates gene expression by enhancing the activity of transcription factors such as NF-κB, SRF, and AP-1 [55]. ASC1 interacts with the promoters of GATA-1 and KLF1 in a UFBP1-dependent manner, emphasizing the interconnected roles of UFBP1, ASC1, and other Ufm1-related components in the survival and differentiation of hematopoietic cells by stabilizing ER function and supporting erythroid gene expression [54]. RCAD/Ufl1, an E3 ligase in the Ufm1 system, regulates DDRGK1, ASC1, ER stress, and autophagy. Its loss disrupts hematopoiesis, causing severe anemia, heightened ER stress, DNA damage, p53 activation, and, ultimately, the death of HSCs [56].

## 3. Pathogenesis of ER Stress and the UPR in AML

Recent studies indicate that AML cells exploit UPR pathways to evade apoptosis, facilitating tumor progression and contributing to their resistance to various therapeutic approaches [21,57,58,59]. The microenvironment in AML influences ER stress levels, affecting disease progression and response to treatment [37]. Leukemic cells interact with their surroundings in ways that can exacerbate ER stress, such as by releasing cytokines and competing for metabolic resources. Mutated proteins predominantly influence the ER stress response in AML; however, it is also affected by genetic alterations and the administration of chemotherapy. Activation of genes such as MYC in AML increases ER tension, which affects various signaling pathways and supports cellular proliferation [60,61]. In the pathogenesis of AML, various oncogenes, including FLT3, NPM1, c-KIT, and RAS, play key roles. Alterations in these genes such as deletions, translocations, inversions, and duplications can keep bone marrow cells from maturing or help cells grow out of control. Uncontrolled profiling enhances protein production, leading to the accumulation of misfolded proteins, ER stress, and activation of the UPR [45,62]. Mutated proteins predominantly influence the ER stress response in AML; however, it is also affected by genetic alterations and the administration of chemotherapy. In NPM1-AML, mutant nucleophosmin mislocalizes in the cytoplasm, disrupting normal nucleolar function and increasing the demand for protein synthesis, making cells more dependent on the UPR for survival. Mutation of NPM-1 causes also diminished sensitivity to cytostatics [63]. Systemic treatment with chemotherapeutic agents used to treat AML, such as cytarabine and daunorubicin, also increases ROS production and ER stress in cells [64,65]. Moreover, new therapies, such as bortezonib (Btz) treatment, prevent the degradation of misfolded proteins, causing them to accumulate in the ER [66].

AML cells exhibit increased glycolysis, resulting in excessive lactate production. This accumulation acidifies the bone marrow microenvironment, disrupting normal hematopoiesis and promoting leukemia progression [67]. Leukemic cells generate high levels of ROS. Conversely, increased levels of ROS generate ER stress [68]. In AML, increased levels of ROS can activate redox-sensitive transcription factors such as NF-κB and AP-1, leading to the higher expression of inflammatory cytokines. Moreover, inflammatory cytokines such as TNF-α and IL-1β can further increase ROS production in leukemic cells, creating a feedback loop that perpetuates oxidative stress and inflammation [69,70].

However, FLT3 wild-type AML cells demonstrate weaker IRE1α, PERK, and ATF6 expression [71]. This suggests lower UPR activation in FLT3 wild-type AML cells. FLT3 mutations play a critical role in driving AML [72]. FLT3-ITD mutations result from duplications within the juxtamembrane region of the receptor. These duplications lead to constitutive, ligand-independent activation of FLT3’s tyrosine kinase activity. The ensuing activation of downstream pathways such as PI3K/AKT, RAS/MAPK, and STAT5 promotes uncontrolled proliferation, survival, and impaired differentiation of hematopoietic progenitors [7,73,74]. Inhibition of IRE1α in FLT3-ITD positive AML cells significantly reduces their clonogenic capacity and induces apoptosis, highlighting the therapeutic potential of targeting ER stress response [57]. In AML cells with FLT3-ITD mutation, the accumulation of the mutated FLT3 protein in the ER disrupts protein processing and alters calcium homeostasis. This disruption impairs calcium transfer to mitochondria, shifting the metabolic balance toward glycolysis, ultimately conferring a survival advantage to AML cells carrying this mutation. Many new therapies are also being developed that target these mutations (Table 1). 

The CEBPA gene plays a vital role in developing myeloid cells and is frequently disrupted in AML. The chaperone calreticulin (CALR) involved in the UPR can bind to a stem-loop structure in the five prime regions of CEBPA mRNA, consequently blocking its translation. This mechanism has been primarily observed in specific AML sub-types characterized by gene rearrangements such as t(3;21) or inv(16). Additionally, the XBP1s were identified in 17.4% of AML patients, and this group exhibited significantly higher expression levels of UPR target genes, including CALR, GRP78, and CHOP [49]. The sustained activation of XBP1s expression in AML cells induces apoptosis both in vitro and in vivo. Moreover, moderate XBP1s expression sensitizes these cells to chemotherapeutic treatments [14]. Furthermore, in vitro studies revealed that calreticulin expression may increase through the ATF6 pathway in myeloid leukemia cells, inhibiting CEBPA protein expression both in vitro and in leukemic cells from patients with an activated UPR [49,75]. 

In myeloid leukemic cells, the overexpression of protein disulfide isomerase (PDI) inhibited the translation of CEBPA without affecting its transcription. Conversely, when PDI was inhibited, the levels of CEBPA protein were restored. It was also discovered that PDI interacts directly with calreticulin. The expression of PDI and a subsequent decrease in CEBPA protein levels were observed when ER stress was induced in these leukemic cells. Notably, 25.4% of AML patients showed signs of ER stress along with elevated levels of PDI. These findings suggest that PDI, as part of the ER stress-associated complex, inhibits the translation of CEBPA and disrupts myeloid differentiation in AML patients experiencing an activated state UPR [76]. The UPR is activated in a significant subset of AML patients through calreticulin induction along the ATF6 pathway, ultimately inhibiting CEBPA translation contributing to the blockade of myeloid differentiation, and contributing to leukemogenesis [77].

Chromatin immunoprecipitation sequencing (ChIP-seq) identified specific target genes of XBP1s, including the long non-coding RNA (lncRNA) MIR22HG, which serves as the precursor transcript of microRNA-22-3p (miR-22-3p). Upregulation of miR-22-3p by XBP1s, or enforced expression of miR-22, significantly decreases cell viability and enhances the sensitivity of leukemic cells to chemotherapy. The intracellular effects of miR-22-3p are mediated, at least in part, by targeting the mRNA encoding the deacetylase sirtuin-1 (SIRT1), a well-established pro-survival factor. Consequently, the novel XBP1s/miR-22/SIRT1 regulatory axis could be pivotal in leukemic cells’ proliferation and chemotherapeutic response [13].

O-GlcNAcylation may aid AML cell survival by modulating the UPR, mainly by inhibiting its pro-apoptotic pathways. Increased levels of O-GlcNAc transferase (OGT) and O-GlcNAc hydrolase (OGA) are associated with heightened expression of UPR-related genes. Moreover, UPR-associated transcription factors may boost the hexosamine biosynthetic pathway activity, promoting O-GlcNAcylation. This process disrupts the PERK-CHOP pathway, reducing apoptosis and supporting AML cell survival under stress conditions, such as hypoxia and proteotoxicity in the bone marrow [78].

**Table 1 ijms-26-03092-t001:** Targeted therapies for the most common genetic mutations in AML.

Target	Compound Name	In Combination	Current Status	Results
FTL3	Gilteritinib	-	FDA-approved	FDA-approved gilteritinib for treatment of adult patients who have relapsed or refractory AML with an FLT3 mutation based on ADMIRAL trial (NCT02421939) [79].
	Quizartinib	Cytarabine + Anthracycline	FDA-approved	FDA-approved quizartinib for treatment of adult patients with newly diagnosed AML that is FLT3 ITD-positive based on QuANTUM-First trial (NCT02668653) [80].
	Midostaurin	Cytarabine + Daunorubicin	FDA-approved	FDA-approved for the treatment of adult patients with newly diagnosed AML with FLT3 mutation [81].
	Lestaurtinib	Cytarabine + Daunorubicin	Phase III	No significant difference was observed in 5-year ORR between the lestaurtinib group (46%) and the control group (45%) [82].
	Sorafenib	Cytarabine + Daunorubicin	Phase III	Sorafenib was well tolerated. Comparison of long-term outcomes suggested that sorafenib exposure was associated with improved event-free survival from study entry as well as disease-free survival and relapse risk from CR but not overall survival [83].
	Crenolanib	-	Phase II	Complete response (CR)/CRi 14,3%, partial response (PR) 16,1%, overall response ratio (ORR) 30.4%, safe, well tolerated. [NCT01657682].
	Sunitinib	Cytarabine + Daunorubicin	Phase II	Sunitinib was well tolerated; 59% of patients achieved CR, 9% achieved partial remission [84].
	Ponatinib	Decitabine + Venetoclax	Phase II	Data unpublished [NCT04188405].
	Tandutinib	-	Phase II	Data unpublished [NCT00297921].
	FF-10101		Phase I	FF-10101 was well tolerated. Composite complete response rate was 10%, and the overall response rate (including partial responses) was 12.5%, including patients who had progressed on gilteritinib [85].
	Cabozantinib	-	Phase I	Data unpublished [NCT01961765].
	Linifanib	Monotherapy in combination with Cytarabine	Phase I	Linifanib was well tolerated. Efficacy was not the primary goal. The rapid disease progression observed in most patients was associated with limited antileukemic activity [86].
IDH1	Ivosidenib	Gilteritinib	Phase I	Data unpublished [NCT05756777].
		Azacitidine	Phase 1b/2	ORR 78.3%, CR 60.9%, safe, well tolerated [87].
IDH2	Enasidenib	-	Phase I	Two-year progression-free 69%, overall survival 74%, safe, well tolerated [88].
			Phase II	Data unpublished [NCT04203316].
		Gilteritinib	Phase I	Data unpublished [NCT05756777].
NPM1	Selinexor	Homoharringtonine + Daunorubicin + Cytarabine or Granulocyte Colony-Stimulating Factor + Aclacinomycin + Cytarabine	Phase III	Data unpublished [NCT05726110].
		Mitoxantrone + Etoposide + Cytarabin	Phase I	Selinexor was well tolerated. The ORR was 43% with 26% CR, 9% Cri, and 9% with a morphologic leukemia-free state [89].
		-	Phase II	Selinexor was well tolerated. Median OS did not differ significantly for selinexor vs. physician’s choice (3.2 vs. 5.6 months) [90].
	Revumenib	Azacitidine + venetoclax	Phase III	Data unpublished [NCT06652438].
		-	Phase I/II	Revumenib was well tolerated. The ORR was 63.2%, CR + CR with partial hematologic recovery 22.8% [91].
	Eltanexor	Venetoclax	Phase I	Data unpublished [NCT06399640].
	Ziftomenib	-	Phase I/II	Data unpublished [NCT04067336].
	DS-1594b	Azacitidine + Venetoclax or Cyclophosphamide + Dexamethasone + Vincristine + Rituximab	Phase I/II	Data unpublished [NCT04752163].
	BMF-219	-	Phase I	Data unpublished [NCT05153330].

## 4. Prognostic Implications of ER Stress Markers in AML

The prognostic significance of ER stress markers in blood cancers has gained increasing attention due to their correlation with treatment outcomes and disease progression. Studies indicate that high levels of the UPR regulator XBP1, a marker of ER stress, correlate with a poor prognosis in hematological malignancies such as MM. Furthermore, XBP1 and IRE1α levels are elevated in MM cells compared to those in healthy individuals [92].

In AML cells, lower levels of CEBPA lead to higher levels of DDIT3. This increase in DDIT3 can boost cell death due to stress in the ER and interfere with the process of forming leukemia cells. The balance between C/EBPα forms, C/EBPα-p30 and C/EBPα-p42, also affects how cells react to chemotherapy. A low ratio of C/EBPα-p42 to C/EBPα-p30 in AML cells is linked to resistance against the BCL2 inhibitor venetoclax [93].

A recent study using RNA-seq data from TCGA-LAML focused on genes related to ER stress and identified 42 genes linked to the prognosis of AML. From this group, a 13-gene risk score model was developed and validated. It includes genes such as MBTPS1, SESN2, BCAP31, CHAC1, CALR, UBE2G2, EIF2AK4, CREB3, AIFM1, WFS1, CTH, PTPN1, and TMTC4. This model effectively classified patients into high- and low-risk groups, with lower risk scores associated with better survival outcomes [24]. Analysis of 483 pediatric AML patients from a children’s oncology group trial showed that low valosin-containing protein (VCP) expression was significantly linked to better five-year survival rates—81% in low VCP patients versus 63% in those with higher levels. VCP is a critical component of ERAD. This association is held regardless of the chemotherapy regimen, with or without Btz. VCP was also an independent predictor of outcomes, with lower VCP levels correlating with higher levels of UPR proteins IRE1 and GRP78. Furthermore, five-year overall survival in patients with low VCP, moderately high IRE1, and high GRP78 improved after treatment with daunorubicin and etoposide combined with Btz compared to daunorubicin and etoposide alone [94]. This may indicate that ER stress neutralizes the drugs included in the gold standard induction regimen "3+7" in AML [94,95].

## 5. Direct Targeting of ER Stress in AML

The potential of IRE1α inhibition, a component of the UPR, by agents such as MKC-3946, 2-hydroxy-1-naphthaldehyde (HNA), STF-083010, and toyocamycin, to disrupt XBP1 mRNA splicing and show cytotoxic effects against AML cells is a promising research area [96,97,98,99,100]. This inhibition triggers caspase-dependent apoptosis and leads to G1 cell cycle arrest, partly regulated by Bcl-2 family proteins, G1 phase control proteins (such as p21cip1, p27kip1, and Cyclin D1), and chaperone proteins [101]. Notably, murine bone marrow cells lacking XBP1 demonstrate resistance to the growth-inhibitory effects of IRE1α inhibitors. Furthermore, they combine HNA with either Btz or arsenic trioxide (AS2O3) to produce synergistic cytotoxicity in AML cells. This synergy is characterized by increased phosphorylation of JNK and decreased phosphorylation of PI3K and MAPK. Additionally, inhibiting IRE1α RNase activity leads to the upregulation of several microRNAs, including miR-34a, in AML cells. In addition, the suppression of miR-34a provides resistance against HNA treatment [75,102].

Moreover, another study indicates that STF-083010 effectively reduced cell survival, inhibited cell growth, and induced apoptosis in AML cell lines and patient samples, all while preserving healthy cells. This selective impact remained consistent even with XBP1 depletion in a mouse model or STF treatment of healthy human cells. Additionally, combining STF-083010 with the FLT3 TKI quizartinib (AC220) significantly increased malignant cell death compared to either treatment alone, indicating a promising dual-target strategy for AML therapy [57]. Further research reveals that combining an IRE1α RNase inhibitor MKC8866 with proteasome inhibitors (Btz and carfilzomib) significantly reduces XBP1s levels. This combination therapy enhances cell death in AML cells, including the resistant CD34+CD38− subpopulation, and impairs their capacity for clonogenic growth [103]. IRE1α inhibitors such as KIRA6 and 4μ8C showed in vitro anti-AML activity in 8 of 18 patient samples, without affecting the viability of healthy donor cells [104]. GSK2656157 selective PERK inhibitor in combination with cytarabine and anthracycline synergistically increases inhibition of AML proliferation in vitro versus each drug alone [105]. The ATF6 pathway is important in the pathogenesis of AML. However, there is a lack of research on ATF6 inhibitors in AML.

## 6. Effect of New Therapy on ER Stress in AML

### 6.1. JUN Inhibition in AML

Inhibiting JUN through short hairpin RNA (shRNA) significantly reduces the viability of AML cells and hampers disease progression in vivo. RNA sequencing analyses indicate that JUN inhibition diminishes the UPR transcriptional activity. Specifically, JUN is activated via MEK signaling in response to ER stress. It binds to the promoters of essential UPR effectors, such as XBP1 and ATF4, stimulating their transcription and assisting AML cells’ edition to ER pressure. Furthermore, shRNA-mediated suppression of XBP1 or ATF4 triggers apoptosis in AML cells. It extends disease latency in vivo, implying that the decreased survival associated with JUN inhibition is linked to the loss of pro-survival UPR signaling [11] (Table 2).

### 6.2. NMT Inhibition in AML

N-myristoylation, a protein modification essential for survival signaling and metabolism, is catalyzed by the enzymes NMT1 and NMT2, which show variable expression in AML cell lines and patient samples. Low NMT2 expression correlates with poor patient outcomes. Zelenirstat, a pioneering pan-NMT inhibitor, has successfully and significantly inhibited myristoylation in AML cells, causing the degradation of Src family kinases, ER stress, apoptosis, and cell death. This successful inhibition underscores the potential of Zelenirstat as a treatment for AML (Table 2). It was tolerated in vivo and reduced leukemic burden in various AML models. LSC-enriched fractions, especially in the OCI-AML22 model, were notably sensitive to myristoylation inhibition. Zelenirstat also disrupted mitochondrial complex I function and oxidative phosphorylation, which is crucial for LSC survival [106].

### 6.3. Proteasome Inhibition in AML

Proteotoxic stress, induced by proteasome inhibition, is emerging as a potential AML treatment strategy. Btz is a selective proteasome inhibitor approved by the FDA for treating MM. It shows efficacy partly by triggering ER stress-dependent apoptosis [107]. In an in vitro study, treatment with proteasome inhibitors such as Btz and carfilzomib increased XBP1s in KG1a and U937 cell lines [103].

AML cells treated with Btz showed downregulation of the m6A regulator WTAP due to increased oxidative stress, as indicated by elevated ROS levels and HMOX-1 activation. In addition, Btz has been linked to ER stress, which further contributes to proteotoxicity. This downregulation can be reversed by the reducing agent N-acetylcysteine. The study also identified modified m6A circRNAs affected by Btz, including circHIPK3, which is involved in protein folding and regulation of oxidative stress [66].

A combination of low doses of the differentiating agent retinoic acid (R), the proteasome inhibitor Btz (B), and the oxidative stress inducer arsenic trioxide (A) shows potent cytotoxic effects on FLT3-ITD+ AML cells by inducing ER stress and oxidative stress (Table 2). However, AML cells gain resistance to RBA in co-culture with bone marrow stromal cells. High-dose ascorbic acid overcomes this resistance, and the RBA-ascorbic acid combination significantly extends survival in a murine FLT3-ITD+ AML model without toxicity. Notably, the treatment disrupts AML-BMSC interactions, causing actin cytoskeleton alterations, increased nuclear thickness, and cytosolic relocalization of YAP in BMSCs [108].

### 6.4. Malachite Green-Mediated Photodynamic Therapy (PDT) in AML

PDT is a therapy that utilizes the cytotoxic properties of visible light in the presence of a photosensitizer and oxygen. A recent study investigated the relationship between PDT’s anti-leukemic effects and ER stress induction in a subtype of AML-acute promyelocytic leukemia cells. The cells were incubated with various concentrations of malachite green and then exposed to specific light conditions. Cell viability was assessed using the MTT assay, and the expression of ER stress markers PERK and GRP78 was evaluated through immunocytochemical staining. The results showed that the combination of malachite green and light significantly decreased cell viability compared to controls treated with malachite green alone, light alone, or neither. Additionally, the treatment group significantly upregulated the expression of PERK and GRP78 [109].

### 6.5. Staphylococcal Enterotoxins in AML

*Staphylococcus aureus* (*S. aureus*) is a common pathogen in patients undergoing therapy for AML. It produces various virulence factors, including Staphylococcal enterotoxins A (SEA) and B (SEB). SEA and SEB significantly alter AML cell behavior. The treatment with SEA and SEB increased AML cell proliferation and resistance to cytarabine while promoting cell migration and invasion, affecting approximately 50% of the cells. Transcriptomic analysis and gene set enrichment analyses, validated by PCR, revealed dysregulation of immune-related genes and pathways. This dysregulation may enable AML cells to evade immune responses and survive in the hostile environment created by these toxins, potentially through the ER stress signaling pathway [110].

### 6.6. Camalexin in AML

Camalexin is a phytoalexin, a naturally occurring antimicrobial compound produced by plants in response to stress, such as pathogen attacks. It inhibits the growth of various cancer cell lines [111]. In AML cells, camalexin induces apoptosis through a caspase-dependent mitochondrial pathway, with ER stress as an upstream regulator. Treatment with camalexin significantly increases the levels of ROS and enhances the activity of superoxide dismutase (SOD) and catalase. Additionally, it raises oxidized glutathione (GSSG) levels while lowering the reduced glutathione (GSH) levels. These changes suggest that ROS generation is crucial for the ER stress and apoptosis triggered by camalexin. In a xenograft mouse model, camalexin effectively suppressed tumor growth without causing noticeable toxicity, indicating that its antitumor effects are mediated through ROS-induced ER stress, leading to mitochondrial apoptosis [112].

### 6.7. PLK4 Inhibition in AML

Polo-like kinase 4 (PLK4) is a crucial regulator of centriole duplication. Abnormal PLK expression is linked to cancer development and progression [113,114]. PLK4 is overexpressed in AML cells. The knockdown of PLK4, or its specific inhibition using centrinone, leads to G2/M phase cell cycle arrest (Table 2). The suppression of cyclin B1 and Cdc2 expression and the promotion of pro-apoptotic protein levels cause this arrest. Furthermore, targeting PLK4 enhances the expression of proteins associated with ER stress, such as GRP78, ATF4, ATF6, and CHOP [115].

### 6.8. Venetoclax in AML

Venetoclax, a drug that targets and inhibits BCL-2, is used to treat AML, often in combination with various agents to enhance therapeutic effects. However, resistance and recurrence remain significant challenges. In vitro and in vivo experiments using AML cell lines, particularly Molm13 and THP-1, have shown that metformin and venetoclax work synergistically to inhibit cell growth and promote apoptosis. The combination of these drugs significantly increases the expression of CHOP, a marker of ER stress. Moreover, reducing CHOP expression through knockdown techniques decreases the apoptosis induced by this drug combination. Additionally, the combination of metformin and venetoclax has demonstrated strong anti-leukemia effects in xenograft models and bone marrow samples from AML patients. These findings suggest that pairing metformin and venetoclax could provide a robust and safe treatment option for AML, warranting further clinical studies [22] (Table 2). A low C/EBPα p42 to p30 ratio in AML cells is linked to resistance against the BCL-2 inhibitor venetoclax, as BCL-2 is a primary target of DDIT3. However, combining venetoclax with ER stress inducers such as tunicamycin and sorafenib can overcome this resistance. These findings indicate that AML patients with a low C/EBPα p42/p30 ratio may not respond well to venetoclax monotherapy but could benefit from combined treatment approaches that include ER stress-inducing agents [93].

### 6.9. PAD Inhibition in AML

Peptidylarginine deiminases (PADs) are a group of enzymes that facilitate post-translational deimination or citrullination of target proteins. This process alters their structure and function and modulates gene regulation [116,117]. Elevated levels of PAD2 and PAD4 have been linked to the progression of acute myeloid leukemia (AML). In vitro studies indicate that the pan-PAD inhibitor BB-Cl-Amidine (BB-Cl-A) induces apoptosis in AML cells, a response not elicited by the PAD4-specific inhibitor. Treatment with BB-Cl-A resulted in the activation of the ER stress response, as evidenced by increased levels of p-PERK and p-eIF2α, ultimately promoting apoptosis in AML cells. The findings advise that PAD2 is essential for maintaining ER homeostasis and mobile survival, indicating that inhibiting PAD2 with BB-Cl-A may also provide a promising therapeutic approach for AML [118] (Table 2).

### 6.10. G Protein-Coupled Estrogen Receptor-1 (GPER) in AML

The GPER has different roles from estrogen receptors ERα and ERβ, and its activation by specific agonists suppresses tumors in various cancers [119]. The proteins S100A8 and S100A9 are stored in neutrophils as a stable heterodimer known as calprotectin. Calprotectin is released in response to infection and activates the innate immune response [120]. S100A9 is extensively overexpressed in AML, and therapeutic strategies aimed toward concentrating on this protein were proven to reduce the viability of AML cells, promoting apoptosis efficiently. This reduction in cell survival is mainly attributed to the downregulation of mTOR and ER pathways.

When S100A9 is silenced, it significantly alters the cellular environment, affecting extracellular acidification and mitochondrial metabolism, essential for energy production and normal cellular function. Furthermore, the S100A9 inhibitor tasquinimod has been shown to primarily disrupt mitochondrial characteristics, highlighting its potential as a targeted therapy.

Significantly, strategies focusing on S100A9 impair the increase of AML cells and enhance their susceptibility to the anti-most cancer agent venetoclax. A high-quality decrease in the anti-apoptotic proteins BCL-2 and c-MYC stages, crucial for most cancer cell survival, characterizes this more advantageous sensitivity. This twin action of S100A9-targeting approaches underscores their capacity as a promising avenue for improving remedy outcomes in AML [121] (Table 2).

Recent research shows that LNS8801, a synthetic agonist of the G protein-coupled estrogen receptor (GPER), selectively induces apoptosis in human AML cells. Studies on AML cell lines and patient samples reveal that LNS8801 inhibits AML cell growth without impacting normal mononuclear cells. Although GPER is present in healthy and malignant myeloid cells, LNS8801’s anti-AML effects appear independent of GPER signaling. Instead, LNS8801 triggers AML cell death mainly through a caspase-dependent apoptosis pathway driven by ROS production and activation of the ER stress response, primarily via IRE1α [122] (Table 2).

### 6.11. N-Acetyltransferase 10 (NAT10) Inhibition in AML

NAT10 is an enzyme with acetyltransferase and RNA binding activities. It serves as a writer enzyme for the N4-acetylcytidine modification of mRNAs. NAT10 is overexpressed in multiple malignancies, including AML, and is linked to poor prognosis, making it a potential therapeutic target [123]. In an in vitro study, NAT10 inhibition—achieved through shRNA knockdown and pharmacological inhibitors—reduced cell proliferation, caused G1 cell cycle arrest, and increased apoptosis in AML cells (Table 2). Mechanistically, NAT10 inhibition reduced CDK2, CDK4, Cyclin D1, and Cyclin E levels while increasing p16 and p21, which are critical for cell cycle regulation. Additionally, NAT10 inhibition induced ER stress by upregulating GRP78 and promoting caspase 12 cleavage, activating the UPR via increased IRE1, CHOP, and PERK levels. This response activated apoptosis by upregulating pro-apoptotic Bax and Bak and downregulating anti-apoptotic Bcl-2 [124].

### 6.12. p53-MDM2 Axis Inhibition in AML

The MDM2 proto-oncogene (MDM2) is a negative regulator of the p53 protein [125]. Dysregulation of MDM2 is associated with poorer outcomes in AML. Studies are ongoing to determine the efficacy of blocking the p53-MDM2 pathway in AML. MDM2 antagonists, such as Nutlin-3a, show limited efficacy alone, prompting the exploration of combination treatments. Nutlin-3a with Triptolid shows a synergistic effect; it inhibits cell proliferation and induces mitochondrial-mediated apoptosis in p53 wild-type (wt) AML cells, in vivo and ex vivo. This combination also reduces tumor growth and leukemia burden in an AML xenograft model (Table 2). Moreover, the combination shows partial effectiveness in some p53-deficient cases. Nutlin-3a enhanced the expression of p53 target genes PUMA and p21, while Triptolide decreased levels of anti-apoptotic factors XIAP and Mcl-1. The combined treatment amplified these effects. Triptolide also operates through p53-independent mechanisms by disrupting the MYC-ATF4 axis, leading to ER stress induction [126].

### 6.13. Lysine Demethylase 6A (KDM6A) in AML

KDM6A is a member of the Jumonji-C (JmjC) domain-containing family of histone demethylases and plays an essential role during embryonic development. Mutations of KDM6A are found in various neoplasms, including MM, acute lymphoid leukemia (ALL), and pancreatic adenocarcinoma [127]. KDM6A regulates immunogenic cell death (ICD) in AML through epigenetic control of genes involved in ER stress and immune signaling pathways. Loss of KDM6A enhances the immunogenicity of AML cells and sensitizes them to immunomodulatory therapies. Targeting KDM6A-deficient AML subtypes with agents that induce ICD or combine epigenetic modulation with immunotherapy could be a promising therapeutic strategy. Moreover, KDM6A deficiency imparts features similar to "BRCAness," making AML cells more sensitive to PARP inhibitors like olaparib. Treatment with olaparib further induced ICD and increased the expression of NKG2D ligands (ULBP1, ULBP5) on AML cells. Combining olaparib with the KDM6A inhibitor, GSK-J4 showed additive effects in promoting ICD and immune activation [128].

### 6.14. ONC201 in AML

ONC201 is a selective antagonist of the dopamine D2 receptor (DRD2) and an allosteric activator of mitochondrial caseinolytic protease P (ClpP). ONC201 also induces ER stress and activates the integrated stress response (ISR) by inducing eIF2α phosphorylation and upregulating ATF4 and CHOP, ultimately promoting apoptotic cell death. In vitro studies show that ONC201 reduces cell viability in multiple malignancies, including AML, ALL, chronic myeloid leukemia (CML), CLL, Hodgkin’s lymphoma, and MM [129].

**Table 2 ijms-26-03092-t002:** New AML therapy affecting ER stress.

Target	Compound Name	In Combination	Current Status	Results
DRD2, ClpP	ONC201	-	Phase I	Data unpublished [NCT03932643].
JUN	shJUN-1 and shJUN-2 knockdown	-	Preclinical phase	Inhibition of XBP1 or ATF4 via shRNA leads to apoptosis in AML cells and significantly prolongs disease latency in vivo, linking reduced survival from JUN inhibition to decreased pro-survival UPR signaling [11].
NMT	Zelenirstat	-	Preclinical phase	Zelenirstat was well tolerated in vivo and caused the inhibition of AML cell lines [106].
Proteasome	MKC8866	Btz or carfilzomib	Preclinical phase	MKC8866, in combination with proteasome inhibitors, significantly reduces XBP1s levels and increases cell death in AML cell lines and patient-derived AML cells [103].
	Btz	-	Phase I	Data unpublished [NCT00077467].
	Retinoic acid	Btz and arsenic trioxide	Preclinical phase	The combination exhibits strong cytotoxic effects on FLT3-ITD+ AML cell lines and primary blasts from patients due to disrupted ER homeostasis and increased oxidative stress [108].
PLK4	Centrinone	-	Preclinical phase	Inhibition of PLK4 induces apoptosis, G2/M, and ER stress in AML cells [115].
BCL-2	Venetoclax	Metformin	Preclinical phase	The combination of metformin and venetoclax showed enhanced anti-leukemic activity with acceptable safety in patients with AML [22].
PAD	BB-Cl-Amidine	-	Preclinical phase	BB-Cl-A activates the ER stress response and, due to this, effectively induces apoptosis in the AML cells [118].
GPER	S100A9 siRNA knockdown	Venetoclax	Preclinical phase	S100A9 knockdown significantly increases venetoclax sensitivity in AML cells [121].
	LNS8801	-	Preclinical phase	LNS8801 induced AML cell death mainly through a caspase-dependent apoptosis pathway, inducing levels of ROS and ER stress response pathways, including IRE1α [122].
NAT10	NAT10 shRNA knockdown	-	Preclinical phase	Targeting NAT10 promotes ER stress, triggers the UPR pathway, and activates the Bax/Bcl-2 axis in AML cells [124].
MDM2	Nutlin-3a	Triptolide	Preclinical phase	The combination exhibits a significant antileukemia effect through both p53-dependent and independent mechanisms, the latter involving the disruption of the MYC-ATF4 axis-mediated ER stress [126].

## 7. Conclusions

Over time, ER stress and the UPR pathway have become established in the pathogenesis of many diseases. However, the role of UPR and ER stress in AML is still poorly understood. Recent research shows that ER stress has a dual role in AML, from promoting leukemic cell survival and resistance to chemotherapy to the possibility of using this pathway in treatment, prognosis, and disease stratification. AML is characterized by high heterogeneity and a large number of genetic variations, which is why treatment is so difficult. The hope for an improved prognosis comes from new therapies that target specific mutations found in AML. A promising example of the development of new treatments is the inhibition of IRE1α in FLT3-ITD+ AML cells. Results of the in vitro studies suggest that targeting ER stress in AML may be a future therapeutic option. Currently, the only therapy targeting ER stress directly is the inhibition of IRE1α. Previous data about IRE1α signaling promoting cancer cell growth and survival in solid tumors and hematopoietic malignancies immediately suggested potential therapeutic vulnerability effects [130,131]. Despite recent advances in available therapies (including targeting mutational diversity), AML remains difficult to treat. Thus, inhibition of IRE1α may selectively affect leukemic cells that rely on its signaling pathways for survival. This dependency highlights the potential of IRE1α as an attractive, non-oncogene addiction, therapeutic target in AML, especially at the secondary stage of the disease (sAML). IRE1α inhibitors have demonstrated effectiveness in inducing apoptosis in AML cells. There is no research with these inhibitors in vivo in animal models or clinical trials with patients. Fieke W. Hoff et al. found that low VCP expression is significantly associated with favorable 5-year OS [94]. Additionally, since IRE1α is negatively correlated with VCP, it may serve as a prognostic and stratification factor. However, there are many therapies where ER stress is involved indirectly. The use of targeted therapy for ER stress in AML appears to be more effective when combined with other drugs rather than as a monotherapeutic treatment. Coming research should focus on the results of IRE1α inhibition in animal models and clinical trials. Additional studies confirming the role of ER stress biomarkers, such as prognostic and stratification factors, are also necessary.

## Figures and Tables

**Figure 1 ijms-26-03092-f001:**
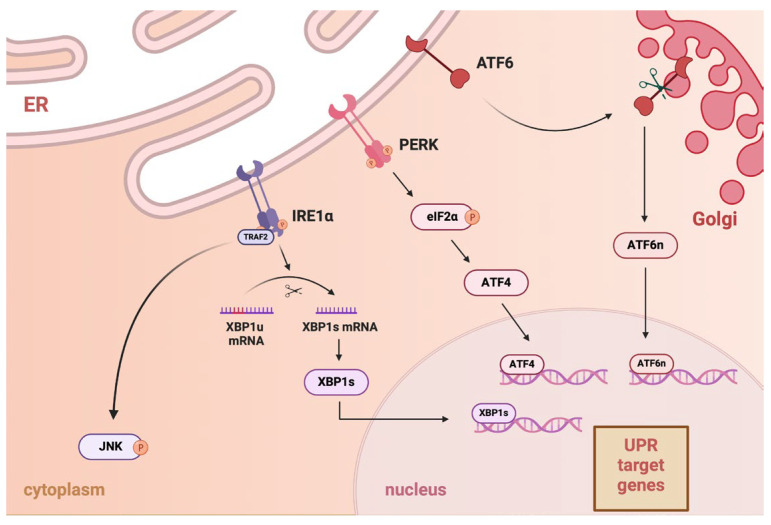
The unfolded protein response (UPR) signaling pathway consists of three major branches: activating transcription factor 6 (ATF6), inositol-requiring enzyme-1α (IRE1α), and PKR-like ER kinase (PERK). During endoplasmic reticulum (ER) stress, they are activated. In the ER, IRE1α binds to TRAF2 and becomes activated. IRE1α splices the XBP1u (unspliced) mRNA to form XBP1s (spliced), then translocated to the nucleus to activate UPR target genes involved in the stress response. IRE1α activates the c-Jun N-terminal kinase (JNK) signaling pathway in the cytoplasm, which may lead to apoptosis. Upon activation, PERK phosphorylates eukaryotic translation initiation factor 2α (eIF2α), which inhibits global protein synthesis. This allows selective translation of ATF4, which then enters the nucleus to activate UPR target genes. ATF6, when activated, moves to the Golgi apparatus, where it is cleaved to form the ATF6n (nuclear form) protein. ATF6n translocates to the nucleus and activates UPR.

**Figure 2 ijms-26-03092-f002:**
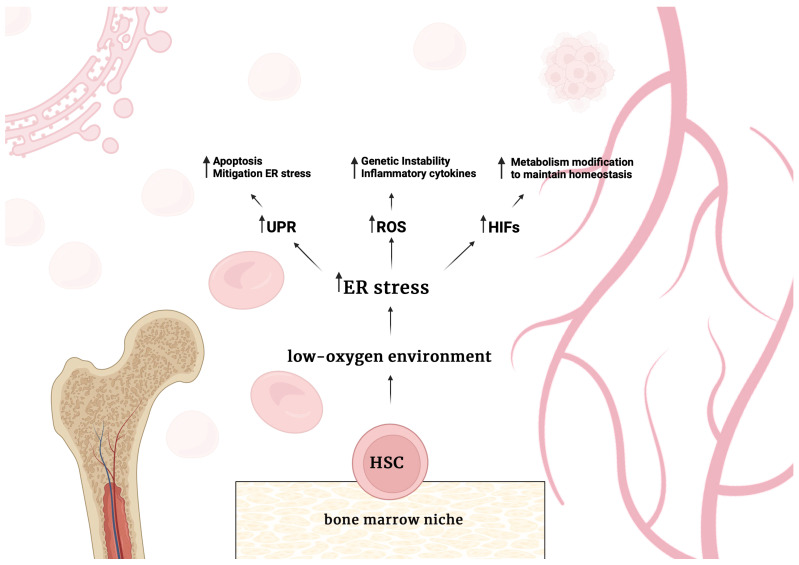
Schematic representation of the main molecular consequences of the conditions that prevail in the bone marrow niche for hematopoietic stem cells (HSCs). A low-oxygen environment increases endoplasmic reticulum (ER) stress levels. As a result, increased ER stress causes an increase in unfolded protein response (UPR) activity, the production of reactive oxygen species (ROS) also increases, and the activity of hypoxia-inducible factors (HIFs).

## Data Availability

The data generated in the present study may be requested from the corresponding author.

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
