# Peer review of "Endoplasmic Reticulum Stress in Acute Myeloid Leukemia: Pathogenesis, Prognostic Implications, and Therapeutic Strategies"

_ijms, 2025, doi:10.3390/ijms26073092_

Round 1
Reviewer 1 Report
Comments and Suggestions for Authors
This is a detailed review providing current knowledge on ER stress and UPR in AML prognosis and new therapeutic approaches in the treatment. Authors have done a thorough literature search from known to recent studies and discussed in detail. This review helps the researchers in the field with up-to-date findings on ER stress in AML and also helps to identify the knowledge gap.
My comments are as follows:
1) Include the discussion of pathogenic importance of ATF6 signaling in different diseases in section 2.
2) Discuss the targeting of the other two signaling pathways PERK and ATF6 in section 5. Authors focused mainly on the IRE1-XBP1 arm of UPR throughout the manuscript, however several recent and previous studies show the importance of the other two signaling pathways.
3) Representative cartoons showing ER stress and UPR with their targets and possible inhibition mechanisms would be helpful.
4) Formatting errors observed at several places.
5) References format should be consistent (eg., 2,33,36)
Author Response
Response to Reviewer 1 Comments
We are grateful to the Reviewer for the care with which our manuscript was read and for the constructive criticism and valuable suggestion. Taking into consideration the suggestions provided, we present an updated version of the article entitled Endoplasmic Reticulum Stress in Acute Myeloid Leukemia: Pathogenesis, Prognostic Implications, and Therapeutic Strategies. Essential changes were made to the text of our manuscript according to the comments and suggestions provided. In the revised version of the manuscript all changes made are highlighted in yellow. We confirm that all authors listed on the manuscript concur with the submission in its revised form. We hope that the revisions in the manuscript and our accompanying responses have made the manuscript suitable for publication in International Journal of Molecular Sciences.
Herein, we explain the revisions that were made based on the comments and recommendations.
Point 1. This is a detailed review providing current knowledge on ER stress and UPR in AML prognosis and new therapeutic approaches in the treatment. Authors have done a thorough literature search from known to recent studies and discussed in detail. This review helps the researchers in the field with up-to-date findings on ER stress in AML and also helps to identify the knowledge gap.
My comments are as follows:
1) Include the discussion of pathogenic importance of ATF6 signaling in different diseases in section
Response 1: We are grateful to the Reviewer for bringing this to our attention. We have enriched the publication with additional information about pathogenic importance of ATF6 signaling in different diseases.
Point 2. Discuss the targeting of the other two signaling pathways PERK and ATF6 in section 5. Authors focused mainly on the IRE1-XBP1 arm of UPR throughout the manuscript, however several recent and previous studies show the importance of the other two signaling pathways.
Response 2: We thank the Reviewer for bringing this to our attention. Based on this suggestion in the revised version of our manuscript we have appended a discussion of the other two PERK and ATF6 signal paths in Section 5.
(line 298)
GSK2656157 selective PERK inhibitor in combination with cytarabine and anthracycline synergistically increases inhibition of AML proliferation in vitro versus each drug alone [100]. ATF6 pathway is important in the pathogenesis of AML. However, there is a lack of research on ATF6 inhibitors in AML.
Point 3. Representative cartoons showing ER stress and UPR with their targets and possible inhibition mechanisms would be helpful.
Response 3: We wish to thank you for your recommendations and appreciation of our figures. In the revised version of our manuscript, we have created a cartoon showing ER stress and UPR with their targets.
Point 4. Formatting errors observed at several places.
Response 4: We thank the Reviewer for this note. In the revised version, the formatting has been corrected.
Point 5. References format should be consistent (eg., 2,33,36)
Response 5: We thank the Reviewer for noticing it. In the revised version, citations have been standardized.

Reviewer 2 Report
Comments and Suggestions for Authors
The manuscript "Endoplasmic Reticulum Stress in Acute Myeloid Leukemia: Pathogenesis, Prognostic Implications, and Therapeutic Strategies" is a manuscript on a relevant topic but it was developed very superficially and requires both a more in-depth bibliographic review and recent references as well as incorporating knowledge that has been acquired in other types of cancer but that can also be applied to AML. For these reasons I consider that the manuscript should be rejected, although I will also make some recommendations.
- Tables and figures do not contribute much. Figure 1, for example, does not contribute anything to the manuscript since it does not describe in depth the molecular mechanisms involved in how the bone marrow environment intervenes to produce ER stress. Similarly, Table 1, in the results section, presents many data marked as "data unpublished", however, there is a large amount of data on cell lines, organoids and in some cases animal models that could be incorporated, for example, Selinexor. In addition, the list of drugs is very limited, for example, for FLT3 there are around 15 drugs in clinical trials.
- It should be mentioned how oncogenes activated by the different mutations associated with AML or their downstream pathways intervene to trigger ER stress. Likewise, silencing or deletion of tumor suppressors.
- Likewise, it should be mentioned how metabolic changes are associated with ER stress, for example, increased glycolysis leading to an increase in lactate and acidity of the hematopoietic microenvironment. ROS and inflammatory cytokines secreted by leukemic cells, etc.
- It should also be mentioned how the drugs included in the 3+7 regimen, the most commonly used in the induction phase (you cite this article 10.1002/prca.202200109 but do not mention it).
- It should also be mentioned how the drugs included in the 3+7 regimen, the most commonly used in the induction phase, are neutralized by ER stresses (you cite this article 10.1002/prca.202200109 but do not mention it). Inhibitors such as GSK2656157 for PERK (or activators) or KIRA6 for IRE1α could be mentioned.
Author Response
Response to Reviewer 2 Comments
We are grateful to the Reviewer for the care with which our manuscript was read and for the constructive criticism and valuable suggestion. Taking into consideration the suggestions provided, we present an updated version of the article entitled Endoplasmic Reticulum Stress in Acute Myeloid Leukemia: Pathogenesis, Prognostic Implications, and Therapeutic Strategies. Essential changes were made to the text of our manuscript according to the comments and suggestions provided. In the revised version of the manuscript all changes made are highlighted in yellow. We confirm that all authors listed on the manuscript concur with the submission in its revised form. We hope that the revisions in the manuscript and our accompanying responses have made the manuscript suitable for publication in International Journal of Molecular Sciences.
Herein, we explain the revisions that were made based on the comments and recommendations.
Point 1. The manuscript "Endoplasmic Reticulum Stress in Acute Myeloid Leukemia: Pathogenesis, Prognostic Implications, and Therapeutic Strategies" is a manuscript on a relevant topic but it was developed very superficially and requires both a more in-depth bibliographic review and recent references as well as incorporating knowledge that has been acquired in other types of cancer but that can also be applied to AML. For these reasons I consider that the manuscript should be rejected, although I will also make some recommendations.
- Tables and figures do not contribute much. Figure 1, for example, does not contribute anything to the manuscript since it does not describe in depth the molecular mechanisms involved in how the bone marrow environment intervenes to produce ER stress. Similarly, Table 1, in the results section, presents many data marked as "data unpublished", however, there is a large amount of data on cell lines, organoids and in some cases animal models that could be incorporated, for example, Selinexor. In addition, the list of drugs is very limited, for example, for FLT3 there are around 15 drugs in clinical trials.
Response 1: We greatly appreciate the Reviewer’s recommendations. We supplemented Figure 1. (now as Figure 2) with additional information in order to make a clear reference to the text of the manuscript.
line (128)
HSCs are exposed to harsh conditions in the bone marrow. The bone marrow niche has low oxygen concentrations, making hematopoietic stem and progenitor cells (HSPCs) susceptible to hypoxia, which can further increase ER stress. In low-oxygen environments, hypoxia-inducible factors (HIFs) activate, mediating a significant adaptive response to hypoxia that involves multiple pathways necessary for maintaining cellular homeostasis. It has been reported that HIF-2α-deficient HSPCs show higher levels of reactive oxygen species (ROS) generation (Figure 2). Elevated ROS can lead to ER stress and trigger apoptosis by activating the UPR. Additionally, ROS can cause genomic instability by damaging DNA, which may contribute to chemotherapy resistance and AML progression [51–53].
As suggested, we have included additional drugs in clinical trials and results to Table 1. We have also supplemented Table 1 with a number of new drugs with careful attention to drugs targeting FLT3.
Point 2. - It should be mentioned how oncogenes activated by the different mutations associated with AML or their downstream pathways intervene to trigger ER stress. Likewise, silencing or deletion of tumor suppressors.
Response 2: We appreciate the Reviewer’s suggestion and agree that it can benefit from additional details. We described how oncogenes and their mutations associated with AML result in increased ER stress.
(line 190)
However, FLT3 wild-type AML cells demonstrate weaker IRE1α, PERK, and ATF6 expression [67]. This suggests lower UPR activation in FLT3 wild-type AML cells. FLT3 mutations play a critical role in driving AML [68]. FLT3‐ITD mutations result from duplications within the juxtamembrane region of the receptor. These duplications lead to constitutive, ligand‐independent activation of FLT3’s tyrosine kinase activity. The ensuing activation of downstream pathways such as PI3K/AKT, RAS/MAPK, and STAT5 promotes uncontrolled proliferation, survival, and impaired differentiation of hematopoietic progenitors [7,69,70]. Inhibition of IRE1α in FLT3‐ITD positive AML cells significantly reduces their clonogenic capacity and induces apoptosis, highlighting the therapeutic potential of targeting ER stress response [56]. In AML cells with FLT3-ITD mutation, the accumulation of the mutated FLT3 protein in the ER disrupts protein processing and alters calcium homeostasis. This disruption impairs calcium transfer to mitochondria, shifting the metabolic balance toward glycolysis, ultimately conferring a survival advantage to AML cells carrying this mutation. Many new therapies are also being developed that target these mutations (Table 1).
Point 3. Likewise, it should be mentioned how metabolic changes are associated with ER stress, for example, increased glycolysis leading to an increase in lactate and acidity of the hematopoietic microenvironment. ROS and inflammatory cytokines secreted by leukemic cells, etc.
Response 3: We thank the Reviewer for this significant comment. In response to these recommendations, we added how metabolic shifts in AML cells are related to ER stress.
(line 182)
AML cells exhibit increased glycolysis, resulting in excessive lactate production. This accumulation acidifies the bone marrow microenvironment, disrupting normal hematopoiesis and promoting leukemia progression [63]. Leukemic cells generate high levels of ROS. Conversely, increased levels of ROS generate ER stress [64]. In AML, increased levels of ROS can activate redox-sensitive transcription factors such as NF-κB and AP-1, leading to the higher expression of inflammatory cytokines. Moreover, inflammatory cytokines such as TNF-α and IL-1β can further increase ROS production in leukemic cells, creating a feedback loop that perpetuates oxidative stress and inflammation [65,66].
Point 4. It should also be mentioned how the drugs included in the 3+7 regimen, the most commonly used in the induction phase (you cite this article 10.1002/prca.202200109 but do not mention it).
- It should also be mentioned how the drugs included in the 3+7 regimen, the most commonly used in the induction phase, are neutralized by ER stresses (you cite this article 10.1002/prca.202200109 but do not mention it). Inhibitors such as GSK2656157 for PERK (or activators) or KIRA6 for IRE1α could be mentioned.
Response 4: We thank the Reviewer for bringing this to our attention. As recommended, we have mentioned that ER stress neutralizes the drugs in the 3+7 regiment. In the revised manuscript, we have also inserted details about IRE1α inhibitors such as KIRA6 and 4μ8 and the PERK inhibitor GSK2656157.
(line 298)
IRE1α inhibitors such as KIRA6 and 4μ8C showed in vitro anti-AML activity in 8 of 18 patient samples, without affecting the viability of healthy donor cells [99]. GSK2656157 selective PERK inhibitor in combination with cytarabine and anthracycline synergistically increases inhibition of AML proliferation in vitro versus each drug alone [100]. ATF6 pathway is important in the pathogenesis of AML. However, there is a lack of research on ATF6 inhibitors in AML.

Reviewer 3 Report
Comments and Suggestions for Authors
In the manuscript „Endoplasmic Reticulum Stress in Acute Myeloid Leukemia: Pathogenesis, Prognostic Implications, and Therapeutic Strategies“ authors summarize the knowledge of the importance of ER stress in the pathogenesis of AML, and how some of the signaling pathways could be targeted. Manuscript is easily readable and reports on the important matter, but since differentiation therapy (ATRA, inhibitors of mutated IDH) is an important part of AML therapy, I would suggest authors to add a section reporting on the role of ER stress in the differentiation of AML.
Author Response
Response to Reviewer 3 Comments
We are grateful to the Reviewer for the care with which our manuscript was read and for the constructive criticism and valuable suggestion. Taking into consideration the suggestions provided, we present an updated version of the article entitled Endoplasmic Reticulum Stress in Acute Myeloid Leukemia: Pathogenesis, Prognostic Implications, and Therapeutic Strategies. Essential changes were made to the text of our manuscript according to the comments and suggestions provided. In the revised version of the manuscript all changes made are highlighted in yellow. We confirm that all authors listed on the manuscript concur with the submission in its revised form. We hope that the revisions in the manuscript and our accompanying responses have made the manuscript suitable for publication in International Journal of Molecular Sciences.
Herein, we explain the revisions that were made based on the comments and recommendations.
Point 1. In the manuscript „Endoplasmic Reticulum Stress in Acute Myeloid Leukemia: Pathogenesis, Prognostic Implications, and Therapeutic Strategies“ authors summarize the knowledge of the importance of ER stress in the pathogenesis of AML, and how some of the signaling pathways could be targeted. Manuscript is easily readable and reports on the important matter, but since differentiation therapy (ATRA, inhibitors of mutated IDH) is an important part of AML therapy, I would suggest authors to add a section reporting on the role of ER stress in the differentiation of AML.
Response 1: We thank the Reviewer for the feedback. We understand the importance of providing detailed information about the role of ER stress in the differentiation of AML. Therefore, in the revised manuscript, we have supplemented the role of ER stress n the differentiation of AML
(line 124)
Activating the UPR pathway blocks myeloid differentiation and deregulates the cell cycle, which are key features of the leukemic phenotype. This occurs by activation of calreticulin in the ATF6 pathway, therefore inhibiting CCAAT/enhancer binding factor alpha (CEBPA), a protein essential for myeloid differentiation [48].

Reviewer 4 Report
Comments and Suggestions for Authors
This review provides a systematic elaboration of the role of ERS in the pathogenesis and treatment of AML. Please allow me to have some comments.
1. Can the authors describe briefly the ASC1 in line 116 briefly and how ERS involves in targeted therapies for NPM1-m AML which was mentioned in in Table 1?
2. ONC201 can also induce ERS in tumor cells and has already been used in the clinical trial for AML.
Author Response
Response to Reviewer 4 Comments
We are grateful to the Reviewer for the care with which our manuscript was read and for the constructive criticism and valuable suggestion. Taking into consideration the suggestions provided, we present an updated version of the article entitled Endoplasmic Reticulum Stress in Acute Myeloid Leukemia: Pathogenesis, Prognostic Implications, and Therapeutic Strategies. Essential changes were made to the text of our manuscript according to the comments and suggestions provided. In the revised version of the manuscript all changes made are highlighted in yellow. We confirm that all authors listed on the manuscript concur with the submission in its revised form. We hope that the revisions in the manuscript and our accompanying responses have made the manuscript suitable for publication in International Journal of Molecular Sciences.
Herein, we explain the revisions that were made based on the comments and recommendations.
Point 1. This review provides a systematic elaboration of the role of ERS in the pathogenesis and treatment of AML. Please allow me to have some comments.
Can the authors describe briefly the ASC1 in line 116 briefly and how ERS involves in targeted therapies for NPM1-m AML which was mentioned in in Table 1?
Response 1: We greatly appreciate the Reviewer’s recommendation. We described ASC1 in the revised manuscript.
(line 151)
ASC1 is a subunit of the ASC-1 complex, a transcriptional coactivator that regulates gene expression by enhancing the activity of transcription factors such as NF-κB, SRF, and AP-1 [56].
(line 174)
In the pathogenesis of AML, oncogenes such as FLT3, NPM1, c-KIT, and RAS. Alterations in these genes such as deletions, translocations, inversions, and duplications can keep bone marrow cells from maturing or help cells grow out of control. Uncontrolled profiling enhances protein production, leading to the accumulation of misfolded proteins, ER stress, and activation of the UPR [44,61]. In NPM1-AML, mutant nucleophosmin mislocalizes in the cytoplasm, disrupting normal nucleolar function and increasing the demand for protein synthesis, making cells more dependent on the UPR for survival. Mutation of NPM-1 causes also diminished sensitivity to cytostatics [62].
Point 2. ONC201 can also induce ERS in tumor cells and has already been used in the clinical trial for AML.
Response 2: We thank the Reviewer for this note. We have addressed your concern by including ONC201 in the revised manuscript.
(line 490)
ONC201 is a selective antagonist of dopamine D2 receptor (DRD2) and an allosteric activator of mitochondrial caseinolytic protease P (ClpP). ONC201 also induces ER stress and activates the integrated stress response (ISR) by inducing eIF2α phosphorylation and upregulating ATF4 and CHOP, ultimately promoting apoptotic cell death. In vitro studies show that ONC201 reduces cell viability in multiple malignancies, including AML, ALL, chronic myeloid leukemia (CML), CLL, Hodgkin's lymphoma, and MM [126].

Reviewer 5 Report
Comments and Suggestions for Authors
The authors provide an overview of ER stress and AML. And include many angels to explain the relevance of ER stress and UPR in AML. There are some concerns:
Lines 139 - 154 - In AML cases with FLT3-ITD mutations, ER stress, and UPR signaling are critical 139 (Table 1). Cells with these mutations rely on UPR signaling to manage the instability of 140 the mutant FLT3 protein - the paper cited states in the abstract that ER stress in reduced in FL3 mutant patients. I'd like this bit re-worked to properly explain UPR with FLT3-ITD.
Also - Table 1 does not fit in where the authors have cited it in the text - expecting a table with FLT3-ITD effects on UPR.
Figure 1 could be built upon to summarise the impact of the up-regulation of UPR, ROS and HIFs.
Author Response
Response to Reviewer 5 Comments
We are grateful to the Reviewer for the care with which our manuscript was read and for the constructive criticism and valuable suggestion. Taking into consideration the suggestions provided, we present an updated version of the article entitled Endoplasmic Reticulum Stress in Acute Myeloid Leukemia: Pathogenesis, Prognostic Implications, and Therapeutic Strategies. Essential changes were made to the text of our manuscript according to the comments and suggestions provided. In the revised version of the manuscript all changes made are highlighted in yellow. We confirm that all authors listed on the manuscript concur with the submission in its revised form. We hope that the revisions in the manuscript and our accompanying responses have made the manuscript suitable for publication in International Journal of Molecular Sciences.
Herein, we explain the revisions that were made based on the comments and recommendations.
Point 1. The authors provide an overview of ER stress and AML. And include many angels to explain the relevance of ER stress and UPR in AML. There are some concerns:
Lines 139 - 154 - In AML cases with FLT3-ITD mutations, ER stress, and UPR signaling are critical 139 (Table 1). Cells with these mutations rely on UPR signaling to manage the instability of 140 the mutant FLT3 protein - the paper cited states in the abstract that ER stress in reduced in FL3 mutant patients. I'd like this bit re-worked to properly explain UPR with FLT3-ITD.
Response 1: We are grateful to the Reviewer for bringing this to our attention. As suggested, we have carefully described FLT3 in the revised manuscript.
(line 190)
However, FLT3 wild-type AML cells demonstrate weaker IRE1α, PERK, and ATF6 expression [67]. This suggests lower UPR activation in FLT3 wild-type AML cells. FLT3 mutations play a critical role in driving AML [68]. FLT3‐ITD mutations result from duplications within the juxtamembrane region of the receptor. These duplications lead to constitutive, ligand‐independent activation of FLT3’s tyrosine kinase activity. The ensuing activation of downstream pathways such as PI3K/AKT, RAS/MAPK, and STAT5 promotes uncontrolled proliferation, survival, and impaired differentiation of hematopoietic progenitors [7,69,70]. Inhibition of IRE1α in FLT3‐ITD positive AML cells significantly reduces their clonogenic capacity and induces apoptosis, highlighting the therapeutic potential of targeting ER stress response [56]. In AML cells with FLT3-ITD mutation, the accumulation of the mutated FLT3 protein in the ER disrupts protein processing and alters calcium homeostasis. This disruption impairs calcium transfer to mitochondria, shifting the metabolic balance toward glycolysis, ultimately conferring a survival advantage to AML cells carrying this mutation. Many new therapies are also being developed that target these mutations (Table 1).
Point 2. Also - Table 1 does not fit in where the authors have cited it in the text - expecting a table with FLT3-ITD effects on UPR.
Response 2: We thank the Reviewer for this comment. In the revised manuscript citation of Table 1 displaced to fit the context more closely.
Point 3. Figure 1 could be built upon to summarise the impact of the up-regulation of UPR, ROS and HIFs.
Response 3: We thank the Reviewer for this comment. As recommended, we have supplemented Figure 1. (now as Figure 2) with the impact of the up-regulation of UPR, ROS and HIFs.

Round 2
Reviewer 2 Report
Comments and Suggestions for Authors
No additional comments
Author Response
We would like to thank the reviewer for his opinion and input on the manuscript.